# Metformin Protects Radiation-Induced Early Brain Injury by Reducing Inflammation and DNA Damage

**DOI:** 10.3390/brainsci13040645

**Published:** 2023-04-10

**Authors:** Jiabao Xiang, Yiming Lu, Cheng Quan, Yan Gao, Gangqiao Zhou

**Affiliations:** 1Graduate Collaborative Training Base of Academy of Military Sciences, Hengyang Medical School, University of South China, Hengyang 421001, China; 2Department of Genetics & Integrative Omics, State Key Laboratory of Proteomics, National Center for Protein Sciences, Beijing Institute of Radiation Medicine, Beijing 100850, China; 3Center of Cognition and Brain Science, Beijing Institute of Basic Medical Sciences, Beijing 100850, China

**Keywords:** metformin, radiation, radiation-induced brain injury

## Abstract

Radiation-induced brain injury (RIBI) is one of the most common and fatal complications of cranial radiation therapy (CRT); however, no effective intervention is available currently. Metformin has been reported to have anti-RIBI activity as a first-line anti-diabetic drug. However, the mechanism of action is unclear. An RIBI mice model and an in vitro cell model under 30 and 10 Gy ^60^Co γ-rays exposure were established to investigate the mechanism of metformin in RIBI. The results showed that pre-treatment with metformin protects hippocampal neurogenesis in the brain of mice and improves learning and memory ability after irradiation. Further investigations revealed that metformin pretreatment reduces inflammation and decreases DNA damage in the in vitro BV2 cell line. In addition, we observed that metformin inhibits the activation of *IκB* and *IRF-3*, which are downstream components of the cGAS-STING pathway. These findings suggest that metformin might protect the brain from RIBI, at least partly, through the cGAS pathway, making it a potential therapeutic drug for RIBI.

## 1. Introduction

Radiotherapy is currently one of the most important adjuvant treatments for several types of head and neck tumors, particularly nasopharyngeal carcinoma [1,2]. Radiation complications are various, and radiation-induced brain injury (RIBI) is known to be the most common side-effect after CRT [2,3], adversely impacting the quality of life and even the survival of patients. A long disease course characterizes RIBI, including brain edema in terms of acute (expressed in days to weeks post-irradiation) and cognitive impairment in late delayed injury (observed >6 months post-irradiation) [4]. Although glucocorticoids and bevacizumab are typically used in clinical therapy for symptoms of RIBI [5], their side effects still cannot be ignored [3]. Therefore, it is urgent to search for safe drug(s) to reduce the adverse symptoms or even prevent the occurrence of IRIB.

Pre-clinical research has provided important insights into possible pathogenic mechanisms connected to RIBI, yet the precise molecular mechanisms or pathways remain mostly unknown. It is increasingly accepted that the pathological process of RIBI is caused by the interaction of astrocytes, endothelial cells, microglia, neurons, and oligodendrocytes in the brain, leading to inflammation, stem cell loss, and disruption of the barrier system [3,6]. Extensive studies have also shown that the accumulation of DNA damage after irradiation may be an important trigger for radiation-induced brain damage [7]. As an evolutionarily conserved abnormal cytoplasmic DNA monitoring system, cGAS/cGAMP/STING is an important part of the innate immune system [8]; however, its role in RIBI’s pathology is unknown. Metformin, a well-tolerated anti-type 2 diabetes medication, is advantageous in promoting brain recovery [9], stimulating hippocampal neurogenesis, and augmenting memory in mice after a brain injury [10,11,12]. Metformin is capable of enhancing both cellular and functional recovery following juvenile cranial radiation [11], indicating that it is a promising candidate for further preclinical and clinical investigations of RIBI. To date, the exact neuroprotective mechanism of metformin remains unknown; thus, further studies are warranted to explore the protective effect and underlying mechanism of metformin on RIBI.

Here, we aimed to explore the protective effect and underlying mechanism of metformin on radiation injury in mouse brain irradiation and cell models. Our study will provide new evidence supporting metformin’s neuroprotective effects on RIBI and novel insights into understanding the underlying mechanisms.

## 2. Materials and Methods

### 2.1. Animal

Male C57BL/6 mice aged 6 to 7 weeks, with a body mass of 20–22 g, were obtained from SPF Biotechnology Co., Ltd. (Beijing, China). These mice were housed up to five mice per cage with a 12 h light-dark cycle at constant temperature (21 °C) and humidity. Access to food and water was ad libitum. All animal experiments in this study were approved by the Institute of Animal Care and Use Committee of the Institute of Radiation Medicine (Beijing, China).

### 2.2. Establishment of RIBI Mouse Models and Drug Administration

All the mice in the radiation treatment groups were irradiated at the Beijing Institute of Radiation Medicine (Beijing, China). Metformin (met; #1396309 Sigma-Aldrich, St. Louis, MO USA) was dissolved in 0.1 M phosphate buffer solution (PBS). The animals were randomly divided into a sham group (sham), metformin treatment group (met), irradiation group (30 Gy), and pre-treatment of metformin irradiation group (30 Gy + met), with ten experimental animals in each group. The mice in met and 30 Gy + met groups were intraperitoneally injected with metformin (20 mg/kg/day) for 7 days, while the sham and 30 Gy groups were given the same amount of PBS. 

For the whole-body model, the mice were irradiated at a dose of 10 Gy γ-rays to define their survival rate. For the whole-brain irradiation model (WBI), the mice were irradiated at a dose of 30 Gy γ-rays at the brain site after anesthetizing with 4% sodium pentobarbital (20 mg/kg) by intraperitoneal injection. The limbs were fixed, and the whole brain was exposed (approximately 0.8 cm from the lower edge of the eye) while using a lead brick to shield the body to protect against γ-rays. The single absorbed dose rate was 90.68 cGy/minute (min), and the distance was 2.5 m (m) from the radiation source. After radiation and drug treatment, we observed the status of mice and recorded their body weight every day to monitor their health status. Mice from the sham group and metformin-treatment group also underwent the same anesthesia procedures as those in the radiation group but did not receive irradiation.

### 2.3. Cell Culture and In Vitro Exposure Models

The mouse microglial BV2 cell line was purchased from Procell Life Science & Technology Co., Ltd. (#CL-0493; Wuhan City, China) and was cultured in Dulbecco’s modified Eagle’s medium (DMEM; Gibco, Invitrogen, Merelbeke, Belgium) containing 10% fetal bovine serum (FBS; Gibco, Invitrogen, Merelbeke, Belgium) and 1% penicillin–streptomycin (Sigma-Aldrich, St. Louis, MO, USA). Cells were maintained in an incubator at 37 °C with 5% CO_2_. To mimic the mouse model of RIBI, BV2 microglial cells were irradiated at a single dose of 10 Gy according to a previous report [4] and subsequently returned to their normal environment (37 °C, 5% CO_2_). The single absorbed dose rate was 88.08 cGy/min, and the distance was 2.5 m from the radiation source. BV2 cells were treated with metformin (50 or 100 μM) [13,14] 12 h before radiation.

### 2.4. Immunofluorescent and Immunocytochemistry Assays

To measure cells proliferation in vivo, thymidine analog 5-Bromo-2-deoxyuridine (BrdU, 20 mg/kg; Sigma-Aldrich, Beverly, MA, USA) was administered intraperitoneally (i.p.) three times per day for 3 days before radiation. Mice were transcardially perfused with PBS (37 °C, pH 7.4), followed by 4% paraformaldehyde. Brains were postfixed overnight in 4% paraformaldehyde at 4 °C and cryoprotected in 30% sucrose for 48 h to dehydrate. Coronal sections (30 mm) of mouse brains were cut using a cryostat (CM3050S; Leica, Wetzlar, Germany) and washed with PBS. For BrdU immunostaining, brain sections were first treated with 1 M hydrochloric acids for 10 min at 4 °C, incubated in 2 M hydrochloric acid for 10 min at room temperature then continued for 20 min at 37 °C, and then rinsed in precooled PBS for 3 times of 10 min. Then, the brain slices were incubated with blocking buffer (0.5% Triton-X-100, 5% BSA in PBS) for 1.5 h at room temperature, followed by incubation with anti-BrdU antibody (1:1000, BU1/75; Abcam, Boston, MA, USA) overnight at 4 °C. The following day, the sections were washed with cold PBS and stained with Alexa FluorR^®^ 488 goat anti-rat IgG secondary antibody (H+L; Invitrogen, Carlsbad, CA, USA) for 1 h. DAPI (300 nM; Solarbio, Beijing, China) was used for nuclear staining. The results were determined by counting the number of BrdU-stained cells compared to non-irradiated controls, based on the total number of BrdU-positive cells in the DG region of each slide from three slides per mouse, with experiments conducted in triplicate.

For γH2AX immunostaining, the BV2 cells, culturing on coverslips coated with poly-L-lysine, were fixed with 4% paraformaldehyde in PBS for 30 min at room temperature. When finished with several washes with PBS, the cells were blocked with PBS containing 5% BSA at room temperature for 1 h. Subsequently, the cells were washed twice with PBS and incubated overnight at 4 °C with rabbit anti-γH2AX primary antibodies (1:5000; Abcam, Boston, MA, USA). Next, after washing three times with PBS, the cells were stained for 2 h at room temperature with a secondary antibody that was conjugated with Alexa FluorR^®^ 546 goat anti-rabbit IgG secondary antibody (H+L; Invitrogen, Carlsbad, CA, USA) diluted at 1:3000 in 3% BSA. The coverslips were then mounted and fixed, and the stained cells were observed using a confocal fluorescence microscope (Nikon 80i; Nikon Corporation, Tokyo, Japan). All the photo documentation was calculated and quantified by ImageJ software v.154d (NIH).

### 2.5. Y-Maze Spontaneous Alternation Test

The Y-maze spontaneous alternation test was used to examine the working memory ability that has been described previously. The maze consists of 3 equally spaced arms (32 cm long, 11 cm wide, and 16 cm high per arm) made of black plastic (designated as arms A, B and C). Mice were placed at the end of one arm of a symmetric Y maze and allowed 5 min to freely explore the apparatus. The series of arm entries are grouped into consecutive sequences of 3 (i.e., ACBCAC = ACB, CBC, BCA, CAC). An alteration was counted when the mouse entered the 3 different arms during a triad on overlapping triplet sets. Spontaneous alternation was measured by counting the number of times mice entered each of the three arms of the maze in succession, divided by the maximum number of possible alternations. The maximum alternations were calculated as follows: the total number of arm entries minus spontaneous alternation (%) = (alternations)/(number of arm entries − 2) × 100.

### 2.6. Quantitative Real-Time Polymerase Chain Reaction (qRT-PCR) Assays

Total RNA from the tissues of mouse brains was extracted, immersed in 200 μL TRIzol (Sigma-Aldrich, Beverly, MA, USA), and broken up with an ultrasonic crusher. After the total RNA was obtained, Superscript III Reverse Transcriptase (Invitrogen, Carlsbad, CA, USA) and random primers were used to synthesize cDNA. After that, the reversed cDNA was amplified using a quantitative PCR reaction system containing 500 ng cDNA, 250 nM upstream and downstream primers, and 12.5 μL of 2 × SYBR Green Realtime PCR Master Mix (TOYOBO, Osaka, Japan). The mRNA levels of all target genes were normalized to the levels of *β-actin*. All primers used in this study are shown in Table 1.

### 2.7. Cell Counting Kit-8 (CCK-8) Assays

The CCK-8 (Vazyme, Nanjing, China) assays were used to detect the radiation’s effect on BV2 cell proliferation and metformin’s toxicity (50 and 100 μM). BV2 cells were seeded (1 × 10^4^ cells per well) into 96-well plates with 100 μL per well. Cells were exposed to 10 Gy radiation and were then treated with 50 or 100 μM metformin. Microplate reader was used to measure absorbance at 450 nm.

### 2.8. Comet Assays

A total of 1 × 10^5^/mL cells was mixed with low-gelling agarose and layered as microgels on microscopic slides. Next, cells were treated overnight with pyrolysis solution at 4 °C and electrophoresed at 23 v 300 mA for 30 min. After staining of gels with SYBR green (Thermo, Carlsbad, CA, USA), results were visualized under a fluorescent microscope. No less than 120 cells were collected for each group to measure their lengths of tails. The level of DNA damage was expressed as the percentage of DNA in the tail. Results were quantified by Comet Assay’s specialized software Open Comet (www.opencomet.org, accessed on 2 November 2022).

### 2.9. Statistical Analyses

All data are expressed as mean ± standard error of the mean (SEM). Statistical analyses were performed using GraphPad Prism software (version 8.0), and the significance of differences was assessed by two-tail unpaired Student’s *t* test or one-way or two-way analysis of variance (ANOVA) followed by Tukey’s multiple comparisons tests. The statistical parameters can be found in the figures and figure legends. *p* < 0.05 was considered to be significantly different.

## 3. Results

### 3.1. Metformin Improves the Survival Time of Mice Subjected to a Lethal Dose of γ-ray Exposure

Metformin has been reported to slow aging and prolong the survival of different cancer patients [9,15,16,17]. In this study, we first evaluated whether metformin affected the survival of mice after radiation injury. Survival curves were constructed for the whole-body irradiation model at a lethal dose of 10 Gy γ-ray irradiation in mice (Figure 1). Before irradiation, mice were injected with metformin intraperitoneally for 7 consecutive days, and their death was recorded every day after irradiation (Figure 1A). We found that all the mice in the irradiation group (10 Gy) were dead within 17 days after radiation, while the survival rate of the metformin group was 52% at this time point (Figure 1B). The results, therefore, indicated that metformin can significantly improve the survival time of mice under a lethal dose of irradiation. However, all mice died within 4 weeks after irradiation, indicating that metformin can only delay but not prevent death in mice.

### 3.2. Metformin Attenuates the Loss of Neural Stem Cells in the Hippocampus and Improves the Learning and Memory Abilities in Mice with RIBI

Memory functions are dependent on the hippocampus during adult neurogenesis in the dentate gyrus (DG) [18]. Previous studies have shown that metformin promotes brain recovery after injury, stimulates hippocampal neurogenesis, and improves memory in mice. Considering that dentate neurogenesis in the adult mouse hippocampus is very sensitive to irradiation, we thus further assessed whether metformin could protect against the loss of neural stem cells of hippocampal DG in the WBI model.

Therefore, a whole-brain irradiation (WBI) model of C57BL/6 mice was established, as shown in Figure 2A. A BrdU-labeled dividing cell was used to assess the proliferation of neural stem cells (NPCs) in the DG (Figure 2B). According to the results, BrdU-positive cells in the hippocampal DG of the 30 Gy group decreased by 80% compared to the sham group (*p* < 0.0001; Figure 2C,D). Although the amount of BrdU-positive cells in the metformin (30 Gy + Met) pre-treatment group was significantly reduced in comparison to the sham–met group, it was still twice as much as the 30 Gy group (*p* < 0.01; Figure 2C,D). Based on these findings, metformin appeared to partially protect neural stem cells in the mice’s hippocampus after irradiation.

We further explored whether metformin treatment protects against cognitive impairment in the WBI model. We used Y-maze spontaneous alternation tests to assess the working memory ability at three months after radiation (Figure 2E,F). The total arm entries and percentage of alternations were calculated to assess the mice’s locomotion and spatial working and memory abilities. First, we found that neither radiation nor pre-treatment metformin affected the locomotion ability of mice (Figure 2G). Then, we found that the alternation rate was significantly reduced in the radiation group compared to the sham group (*p* < 0.01; Figure 2H), and the metformin treatment group had a 17% increase in the alternation rate compared with the radiation group (*p* < 0.01; Figure 2H), suggesting that the metformin treatment significantly reverses radiation-induced decline in working memory ability. These results suggested that pre-treatment of metformin in the WBI model can ameliorate γ-ray-induced cognitive decline three months after radiation.

### 3.3. Metformin Reduces the Inflammatory Response to Whole-Brain Irradiation in Mice

Pieces of evidence have shown that irradiation-induced neuro-inflammation is the most common side effect of RIBI in the acute phase [19]. Here, we measured the expression levels of IL-6, IL-1β and TNF-α in mice’s whole brain (WB) at the acute phase of WBI (Figure 3A). The qRT-PCR assays showed that compared with the sham group, the mRNA levels of IL-6, IL-1β and TNF-α had increased by roughly twice at the 6th and 24th h after γ-ray irradiation (Figure 3B–D). Nonetheless, pre-treatment with metformin significantly reduces the expression levels of these pro-inflammatory cytokines; thus, these three pro-inflammatory cytokines decreased to the level of the sham group (Figure 3B–D), indicating that metformin can effectively suppress acute brain inflammation induced by γ-ray irradiation.

We further isolated the hippocampus and cortex to evaluate the expression changes of the inflammatory factors. The qRT-PCR assays showed that after γ-ray irradiation, the mRNA levels of these three pro-inflammatory cytokines increased by two times compared with the sham group at the 6th and 24th h in both the hippocampus and cortex (Figure 3B–D), while pre-treatment with metformin significantly reduces the expression levels of these pro-inflammatory cytokines to the level of the sham group (Figure 3B–D). These findings are in agreement with the trend of those cytokines in the whole brain, indicating that pre-treatment with metformin decreases the expression of anti-inflammation in both whole and subregions. These findings indicate that metformin could effectively inhibit the cerebral inflammatory response in the RIBI mice without regional brain specificity.

### 3.4. Metformin Reduces the Inflammatory Response of BV2 Cells after Irradiation

It was known that increased inflammatory cytokines after radiation are mainly attributed to microglia [20]. Increasing evidence has shown that microglia activation-mediated inflammation is a major factor in neuronal injury, which contributes significantly to the development of neuroinflammation and various brain diseases. Therefore, these findings suggest that the protective effect of metformin against inflammation in RIBI may be related to microglia. In order to understand how metformin protects against ionizing irradiation injury, we investigated it using BV2 cells.

We first evaluated the toxic effects of metformin on BV2 cells using CCK-8 assay (Figure 4A). The findings revealed that pre-treatment with metformin (50 and 100 μM) did not affect the proliferative activity of BV2 cells. As a next step, we evaluated the effect of metformin on the proliferation of BV2 cells after irradiation (Figure 4B). The results show that 6 and 24 h after 10 Gy irradiation, the proliferation activity of the BV2 cells was reduced to 70% of the sham group (*p* < 0.0001), while both the 50 and 100 μM doses of metformin pre-treatment could restore the proliferation activity of BV2 cell after irradiation by about 18% compared to the irradiation group (Figure 4C,D). These results showed that pre-treatment of metformin has a protective role on the proliferative activity of BV2 cells after irradiation.

Then, we investigated the levels of IL-6, IL-1β and TNF-α responding to metformin treatment in the BV2 cell irradiation model (Figure 4E). At the six-hour time-point after 10 Gy irradiation, the three inflammatory factors were upregulated by two or more times (Figure 4F); however, both concentrations of metformin (50 and 100 μM) reduced their expression. This phenomenon is more obvious 24 h after irradiation (Figure 4G). Additionally, both 50 and 100 μM concentrations of metformin could significantly reduce the increases in these three inflammatory factors induced by irradiation. Together, these results indicate that metformin could suppress the production of inflammatory cytokines in response to 10 Gy irradiation.

### 3.5. Metformin Reduced DNA Damage in BV2 Cells after Irradiation

It has been shown in the literature that DNA damage is one of the initial causative factors in the inflammation process in RIBI, which further leads to a reduction in the proliferative capacity of vascular endothelial or brain glial cells and causes progressive and irreversible brain damage [6]. Thus, we wondered whether metformin exerted neuroprotective effects against DNA damage in RIBI. It was well known that γH2AX can mark DNA damage sites [7]. First, we examined the effects of metformin on the expression of γH2AX early after irradiation. We found a 35% increase in the number of γH2AX-positive cells 1 h after 10 Gy γ-ray irradiation compared with the sham group (*p* < 0.0001; Figure 5A,B), while that rate decreased to 5% in the metformin-treated group (*p* < 0.0001; Figure 5A,B). In addition, comet assays were used to measure DNA damage (Figure 5C). Since the amount of DNA double-strand breaks is overall proportional to the amount of DNA in the tail compared to the DNA remaining in the head [21], the DNA damage level was quantified as the percentage of DNA in the tail (tail DNA%) and tail moment. The data demonstrated that tail DNA% and tail moment in irradiation groups were significantly increased after 10 Gy radiation when compared to the sham groups (*p* < 0.0001), whereas these values were significantly lower in the metformin pretreatment groups of both 50 and 100 μM groups in comparison to the irradiation groups (*p* < 0.0001; Figure 5D,E). Based on these findings, metformin reduced DNA damage caused by irradiation in BV2 cells.

### 3.6. Metformin Reduced the Activation of cGAS-STING after Irradiation

cGAS/cGAMP/STING is an evolutionarily conserved abnormal cytoplasmic DNA monitoring system that plays an important role in the innate immune system [22,23]. It is closely associated with radiation resistance to tumor irradiation therapy [24,25]. Thus, we further explored whether the cGAS/STING pathway participates in the protective effects of metformin in RIBI. The expression changes of *IκB* and *IRF-3*, two downstream factors of the STING pathway, were examined. The results showed that compared with the sham group, the mRNA expression levels of *IκB* and *IRF-3* were increased by nearly two times at the 6th h after 10 Gy γ-irradiation (Figure 6A,B). These two factors increased by about 1.5 times at the 24th h after radiation (Figure 6C,D), indicating that the STING pathway was activated after γ-irradiation. In contrast, the mRNA expression levels of *IκB* and *IRF-3* were significantly reduced in the metformin-treated group after irradiation (Figure 6A–D). It was found that metformin reduced the activation of the cGAS-STING pathway, which contributes to DNA damage and inflammation.

## 4. Discussion

There are different degrees of RIBI associated with radiotherapy for patients with brain tumors, which can result in neuroinflammation, brain tissue injury, cognitive impairment, and necrosis in the brain [3,5]. Although glucocorticoids and bevacizumab have been used in clinical treatment for RIBI, the current treatments restrict symptomatic therapy and have disadvantages such as inconstant transient efficacy, sequelae, and complications [3]. In this study, we verified that pre-administration of metformin prolonged the survival time of mice under a lethal dose of whole-body irradiation exposure. We further found that in the whole-brain radiation model, pre-treatment with metformin effectively inhibits radiation-induced neuro-inflammation, reduces the loss of neural stem cells in the DG region of the hippocampus, and improves cognitive impairment. We also revealed that metformin could effectively inhibit the inflammatory response and DNA damage in the radiation model of BV2 cells. The protective effect may be produced by inhibiting the activation of *IκB* and *IRF-3*. Therefore, our preliminary findings uncovered the potential mechanisms of metformin protecting RIBI and provided a preventive strategy for RIBI.

Metformin has been widely used as a clinical drug for treating type II diabetes for decades. In recent years, studies have shown that besides type II diabetes, metformin also increases the radiosensitivity of various cancers [26,27] and improves the effect of radiotherapy through the NF-κB pathway [28,29,30]. Metformin has been shown to promote neurogenesis, myelination, synaptic pruning, and even promote functional recovery in children with RIBI [31]. Given that metformin is efficacious and safe in various diseases, this study explored the protective and possible protective effects of metformin on RIBI in animal and cell models. Previous studies have suggested that the early damaging effects of irradiation drove cognitive impairment in the late-response phase of radiation brain injury [3]. Therefore, a narrow window of time before or after irradiation is an important time window for the treatment of radiation-induced brain injury to reduce the multiple adverse effects caused by radiation. This study revealed that the pretreatment with metformin one week before radiation not only significantly attenuated the inflammatory response during the acute phase but also ameliorated cognitive impairment. This implies that the one-week period before irradiation could be a promising therapeutic window for RIBI. The significance of these findings lies in the potential to establish new RIBI interventional strategies and to advance metformin’s clinical use in treating RIBI.

Metformin’s ability to inhibit the inflammatory response may be the underlying mechanism behind its neuroprotective effect on neural stem cells, ultimately protecting cognitive damage in RIBI. An important factor related to cognitive impairment is microglia-mediated inflammation [32,33]. Decreased hippocampal neurogenesis is thought to be responsible for memory function decline following brain radiation therapy. The activation of microglia releases proinflammatory factors and produces various neurotoxic factors, which directly lead to the decrease in hippocampal neurogenesis [34], inhibition of long-term potentiation, decreased synaptic function, and even neuronal death [35]. Therefore, the inflammatory response is not only the pathological mechanism of cognitive impairment caused by RIBI but is also an important intervention target used to improve RIBI.

Nevertheless, the intricate mechanism by which metformin influences microglia to ameliorate RIBI injury is still to be revealed. Although the neuroinflammatory response is the most important way for microglia to respond to external stimuli, it is not universal in all situations. There is morphological diversity among microglia in development, health, and injury [36], and different functions are often accompanied by regional differences in microglial distribution and morphology. With the application of single-cell RNA sequencing in a variety of brain diseases, including autism, brain tumors, and neurodegenerative disorders, studies have found that different types of neurodegenerative diseases are mediated by different subtypes of microglia [36,37,38], rather than by changes in the number or morphology of glia. Based on these findings, therefore, future investigations will employ single-cell sequencing technology to identify microglial subpopulations that are sensitive to RIBI and those that are targeted by metformin. In addition, rescue or intervention experiments are essential to gain further insight into the molecular mechanism responsible for metformin’s protective effect on RIBI.

## 5. Conclusions

The objectives of this study were to examine possible mechanisms underlying the protective effects of metformin against RIBI in vivo and in vitro. It is undeniable that metformin can protect cognitive function, suppress the inflammatory response brought on by RIBI, and lessen the loss of nerve cells brought on by ionizing radiation. Metformin’s ability to reduce DNA damage and the inflammatory response to radiation is further demonstrated at the cellular level by the drug’s ability to block the cGAS pathway. This investigation has deepened our knowledge of metformin’s effects and has supplied evidence for its application in radiotherapy.

## Figures and Tables

**Figure 1 brainsci-13-00645-f001:**
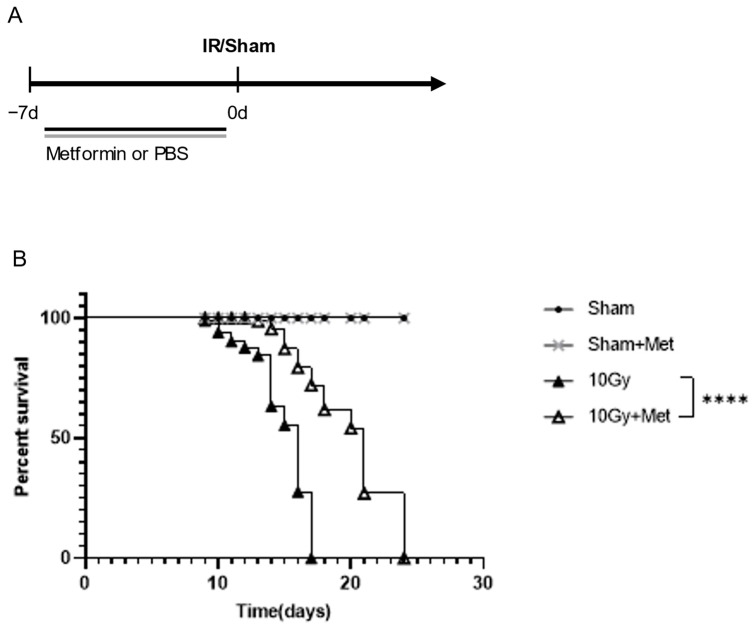
Metformin can reduce the mortality rate in mice exposed to γ-irradiation. (**A**) Overview of experimental timeline. IR, irradiation; PBS, phosphate buffer solution; d, day. (**B**) Survival curves for the mice by whole body exposure of 10 Gy γ-irradiation (n = 17). Met, metformin. **** *p* < 0.0001.

**Figure 2 brainsci-13-00645-f002:**
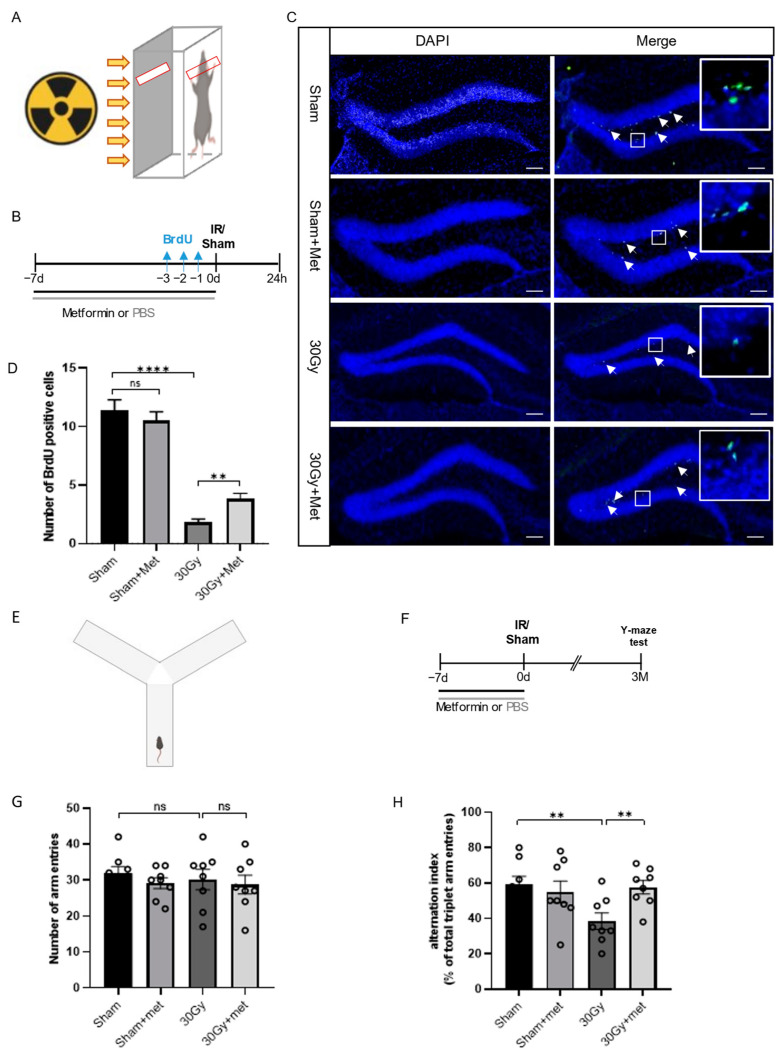
Metformin reduced the loss of neural stem cells and improved working memory scores and behavior in the acute phase of the whole-brain irradiation (WBI) mice model. (**A**) Schematic illustration of the irradiation exposure setup of mice. (**B**) Overview of the experimental timeline. IR, irradiation; PBS, phosphate buffer solution; d, day; h, hour. (**C**) Representative immunofluorescence visual field map. Met, metformin. (**D**) The number of BrdU-positive cells in the hippocampus of mice (30 Gy, n = 12). (**E**) Schematic of the Y maze test. (**F**) Overview of the experimental timeline. m, month. (**G**) The number of total arm entries in four different groups (30 Gy, n = 8/group). (**H**) The effects of metformin on the alternation index (% of total triplet arm entries) (30 Gy, n = 8/group). ** *p* < 0.01, **** *p* < 0.0001, n.s., not significant.

**Figure 3 brainsci-13-00645-f003:**
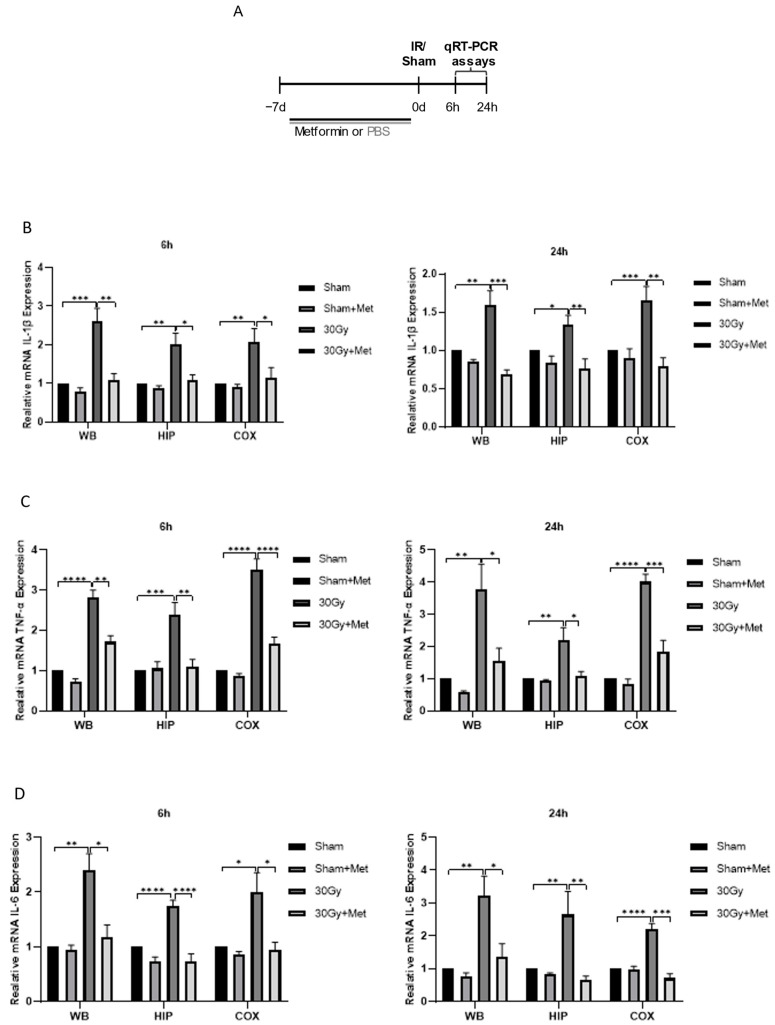
The inhibitory effect of metformin on acute inflammation in mice of the whole-brain irradiation model. (**A**) Overview of the experimental timeline. IR, irradiation; PBS, phosphate buffer solution; d, day; h, hour. (**B**–**D**) The mRNA expression levels of IL-1β (**B**), TNF-α (**C**) and IL-6 (**D**) at the 6th and 24th h after irradiation (30 Gy, n ≥ 7/group). Met, metformin; WB, the whole brain; HIP, the hippocampus; COX, cortex. * *p* < 0.05, ** *p* < 0.01, *** *p* < 0.001, **** *p* < 0.0001.

**Figure 4 brainsci-13-00645-f004:**
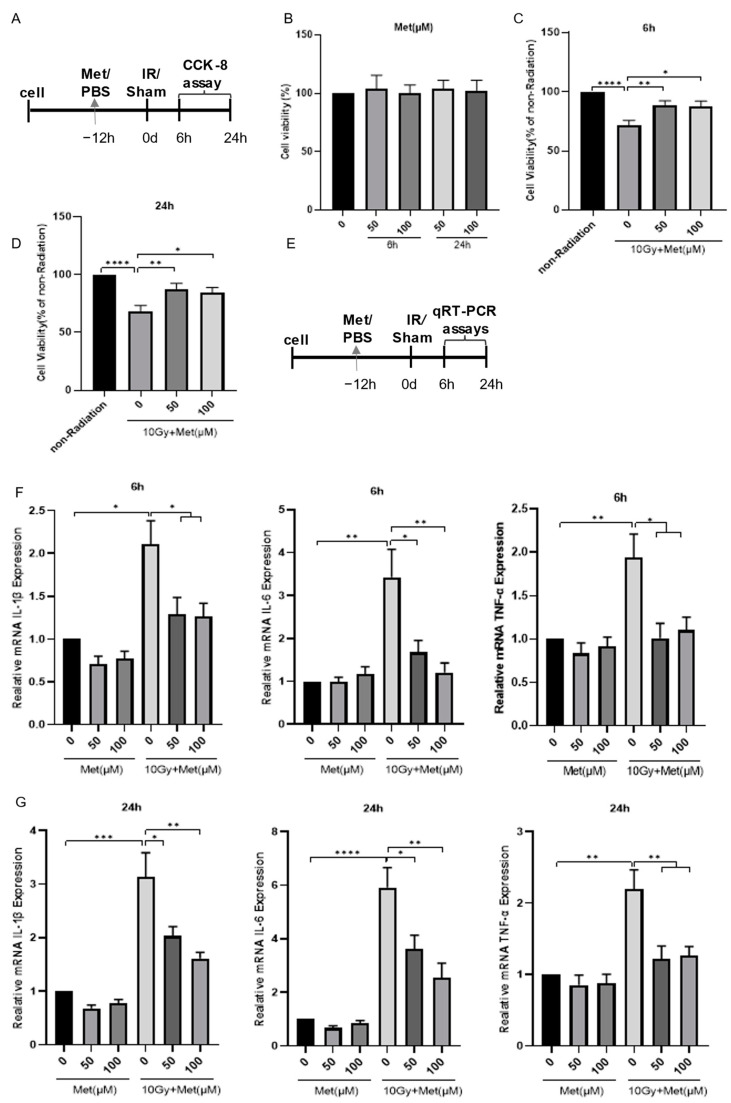
Metformin reduces radiation-induced inflammation in BV2 cells. (**A**) Overview of the CCK-8 assay timeline. Met, metformin; PBS, phosphate buffer solution; IR, irradiation; h, hour. (**B**) The effect of metformin on the viability of BV2 cells was detected by CCK-8 assays. (**C**,**D**) The effect of metformin on the proliferation viability of BV2 cells was detected by CCK-8 assays 6 (**C**) and 24 h (**D**) after 10 Gy irradiation. (**E**) Schematic diagram of the in vitro irradiation model (10 Gy). (**F**,**G**) The mRNA expression levels of IL-1β, TNF-α and IL-6 at the 6th (**F**) and the 24th h (**G**) after irradiation. * *p* < 0.05, ** *p* < 0.01, *** *p* < 0.001, **** *p* < 0.0001.

**Figure 5 brainsci-13-00645-f005:**
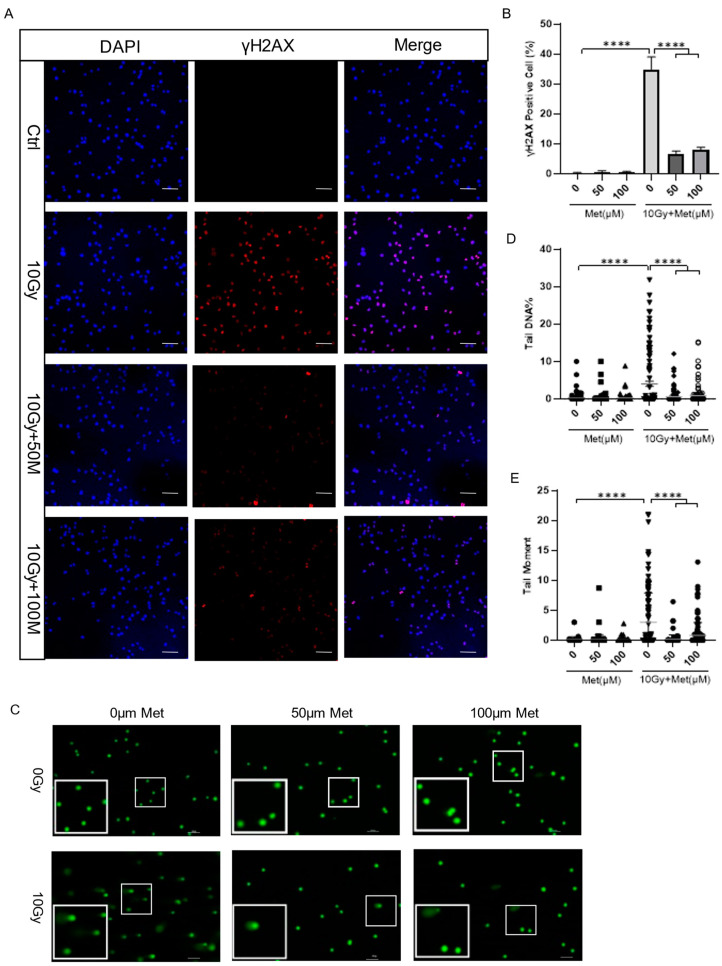
Metformin reduces DNA damage in BV2 cells by γ-H2AX immunostaining assays and comet assays. (**A**) Representative immunofluorescence visual field map in BV2 cells after 10 Gy irradiation. (**B**) The quantitative data of the number of γH2AX-positive cells in immunofluorescence after irradiation are expressed as the ratio of the number of γH2AX-positive cells to the number of DAPI-positive cells in the visual field. Met, metformin. (**C**) Representative images of comet electrophoresis in BV2 cells after 10 Gy irradiation. (**D**,**E**) Quantitative analysis and statistics of percentage of DNA in the tail (tail DNA%) (**D**) and tail moment (**E**) in the comet electrophoresis experiment. **** *p* < 0.0001.

**Figure 6 brainsci-13-00645-f006:**
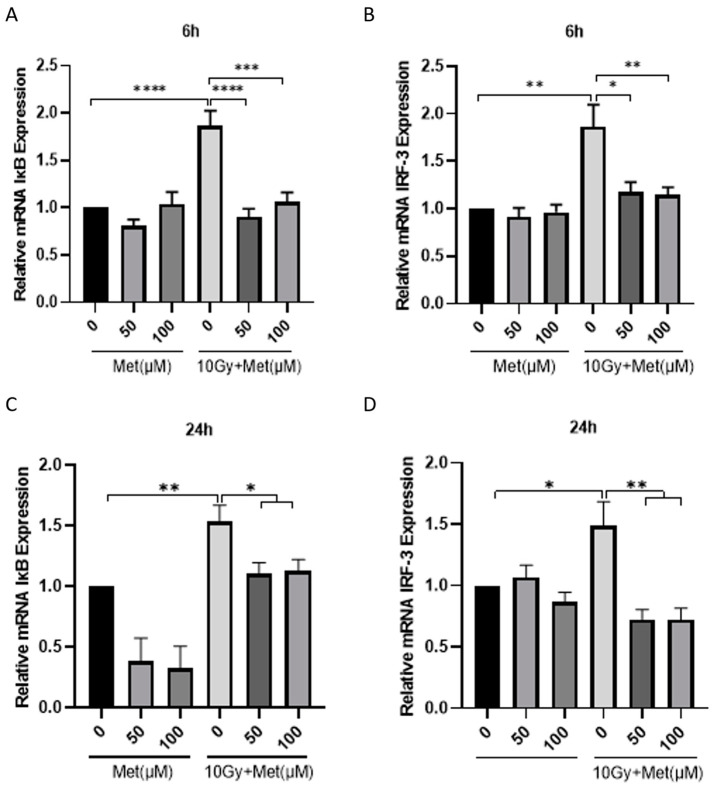
Metformin reduces the mRNA levels of *IκB* and *IRF-3* in BV2 cells. (**A**,**B**) The mRNA expression levels of *IκB* (**A**) and *IRF-3* (**B**) at the 6th hour (h) after 10 Gy irradiation. (**C**,**D**) The mRNA expression levels of *IκB* (**C**) and *IRF-3* (**D**) at the 24th h after 10 Gy irradiation. Met, metformin. * *p* < 0.05, ** *p* < 0.01, *** *p* < 0.001, **** *p* < 0.0001.

**Table 1 brainsci-13-00645-t001:** Primers used for qRT-PCR assays.

Genes	Primers	Primer Sequences (5′→3′)
*IL-1β*	Forward	TGTAATGAAAGACGGCACACC
	Reverse	TCTTCTTTGGGTATTGCTTGG
*IL-6*	Forward	TCCAGTTGCCTTCTTGGGAC′
	Reverse	GTGTAATTAAGCCTCCGACTTG
*TNF-α*	Forward	CAGGCGGTGCCTATGTCTC
	Reverse	CGATCACqCCCGAAGTTCAGTAG
*IκB*	Forward	CAGCTCCAAGAAAGGACGAAC
	Reverse	GGCAGTGTAACTCTTCTGCAT
*IRF-3*	Forward	GAGAGCCGAACGAGGTTCAG
	Reverse	CTTCCAGGTTGACACGTCCG
*Actin*	Forward	ATCGTACTCCTGCTTGCTGAT
	Reverse	AGATTACTGCTCTGGCTCCTAG

## Data Availability

Data are available from the authors on request.

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
