# Peer review of "Metformin Protects Radiation-Induced Early Brain Injury by Reducing Inflammation and DNA Damage"

_brainsci, 2023, doi:10.3390/brainsci13040645_

Round 1
Reviewer 1 Report
The paper entitled "Metformin protects radiation-induced early brain injury by reducing inflammation and DNA damage" is well structured, the authors explored the protective effect and underlying mechanism of metformin on radiation injury in the mouse and cell model.
The authors explain that microglia-mediated inflammation is considered to be a possible mechanism underlying the occurrence or deterioration of RIBI, they could integrate the histological section demonstrating a reduction of glial cells response in treated mice using antibody anti-glial cells like GFAP or MAC 307 for example.
I suggest this experimental improvement to make the results even stronger.
Author Response
Response to Reviewer 1 Comments
Dear Reviewer,
Thanks very much for taking the time to review and for these precious comments concerning my manuscript entitled “Metformin protects radiation-induced early brain injury by reducing inflammation and DNA damage” (Manuscript ID: brainsci-2252624). These comments are all valuable and very helpful for revising and improving my paper, as well as the important guiding significance to my research According to your advice, we amended the relevant part in our manuscript. The followings are our point-to-point responses to your comments.
Point 1: English language and style are fine/minor spell check is required.
Response 1: We apologize for the language problems in the original manuscript. The language presentation was improved with assistance from a native English speaker with an appropriate research background.
Point 2: Check that all references are relevant to the contents of the manuscript.
Response 2: We sincerely appreciate the valuable comments. We have checked the literature carefully and deleted some references that were inappropriate or wrongly cited in the original file and updated it with Endnote.
Point 3: The authors explain that microglia-mediated inflammation is considered to be a possible mechanism underlying the occurrence or deterioration of RIBI, they could integrate the histological section demonstrating a reduction of glial cells response in treated mice using antibody anti-glial cells like GFAP or MAC 307 for example.
I suggest this experimental improvement to make the results even stronger.
Response 3: We are appreciative of your suggestion. Indeed, it will be more profound if we get the result of metformin targeting mice glial cells.
Using previously collected brain slices, we employed immunofluorescence to measure changes in microglial (IBA1) and astrocytes (GFAP) numbers in the DG region of the hippocampus of mice 1 week after radiation treatment. Unfortunately, results are unavailable at this point. Discernible variations in the number of glial cells across the groups have yet to be discovered in our results. One study showed that elevated GFAP levels were increased between 120 and 180 days after single doses of 20-45 Gy radiation [1]. The accumulation of microglial cells occurs early (6 h or 24 h) after whole brain irradiation, but returns toward normal within 1 week [2]. These results are consistent with our results. Moreover, studies indicate that different functions are often accompanied by regional differences in microglial distribution and morphology. With the application of single-cell RNA sequencing in a variety of brain diseases, including autism, brain tumors, and neurodegenerative disorders, studies have found that different types of neurodegenerative diseases are mediated by different subtypes of microglia [3-5], not due to changes in the number or simple morphology of glia.
This article focuses on the preventive effect of metformin on the RIBI model. In our ongoing study, RIBI-sensitive microglia subsets have been identified using single-cell sequencing (unpublished). Therefore, in follow-up studies, we can further investigate the effect of metformin on RIBI-sensitive subsets and explore the role and molecular mechanism of metformin on RIBI.
We also add this part to the “Discussion” section for explanation. We seek for your tolerance and understanding. Many thanks for your kind help!
[1] CHIANG C S, MCBRIDE W H, WITHERS H R. Radiation-induced astrocytic and microglial responses in mouse brain [J]. Radiother Oncol, 1993, 29(1): 60-68.
[2] HAN W, UMEKAWA T, ZHOU K, et al. Cranial irradiation induces transient microglia accumulation, followed by long-lasting inflammation and loss of microglia [J]. Oncotarget, 2016, 7(50): 82305-82323.
[3] HAMMOND T R, DUFORT C, DISSING-OLESEN L, et al. Single-Cell RNA Sequencing of Microglia throughout the Mouse Lifespan and in the Injured Brain Reveals Complex Cell-State Changes [J]. Immunity, 2019, 50(1): 253-271 e256.
[4] OLAH M, MENON V, HABIB N, et al. Single cell RNA sequencing of human microglia uncovers a subset associated with Alzheimer's disease [J]. Nat Commun, 2020, 11(1): 6129.
[5] SMAJIC S, PRADA-MEDINA C A, LANDOULSI Z, et al. Single-cell sequencing of human midbrain reveals glial activation and a Parkinson-specific neuronal state [J]. Brain, 2022, 145(3): 964-978.
Point 4: Are the conclusions supported by the results? Can be improved
Response 4: Thanks for your comment. The manuscript has added relevant information in the “Discussion” section and rewritten the “Conclusion”.

Reviewer 2 Report
This research articles interesting and well presented, and provides useful information about the role of Metformin against radiation-induced early brain injury by reducing inflammation and DNA damage. The manuscript is written well and acceptable for publication but need minor revision. 1) Manuscript language can be improved. 2) Discussion part can be improved. 3) The manuscript needs minor correction, spelling, spacing etc. 4) Endnote should be used for citation and reference.
Author Response
Point 1: Manuscript language can be improved.
Response 1: Thanks for your suggestion. We feel sorry for our poor writing. The English usage and syntax have been polished throughout the manuscript by Wiley Language Editing Services. The certification was also attached.
Point 2: Discussion part can be improved.
Response 2: Thanks for your comment. The manuscript has added relevant information in the “Discussion” section and rewritten the “Conclusion”.
Point 3: The manuscript needs minor corrections, spelling, spacing, etc.
Response 3:We are very sorry for the mistakes in this manuscript and the inconvenience they caused in your reading. Several inaccuracies have been corrected after reading the manuscript several times, and the corrections are evident under *Track Changes*.
Point 4: Endnote should be used for citation and reference.
Response: We sincerely appreciate the valuable comments. We carefully reviewed the literature, updated it with Endnote, and eliminated any references that were incomplete or incorrectly mentioned in the original file.

Round 2
Reviewer 1 Report
The authors improved the paper according to suggestions and checked the English language.
The paper could be accepted in the present form
Author Response
Thank you for your suggestion.